# Overcoming Tribal Boundaries: The Biocultural Heritage of Foraging and Cooking Wild Vegetables among Four Pathan Groups in the Gadoon Valley, NW Pakistan

**DOI:** 10.3390/biology10060537

**Published:** 2021-06-15

**Authors:** Sheharyar Khan, Wahid Hussain, Sikandar Shah, Hidayat Hussain, Ahmed E. Altyar, Mohamed L. Ashour, Andrea Pieroni

**Affiliations:** 1Department of Botany, University of Peshawar, Peshawar 25120, KP, Pakistan; sheharyarbotany@uop.edu.pk (S.K.); sulaiman097@uop.edu.pk (S.); sikandarbotanist@uop.edu.pk (S.S.); 2Department of Botany, Government Post Graduate College, Parachinar 26000, KP, Pakistan; 3Department of Bioorganic Chemistry, Leibniz Institute of Plant Biochemistry, 6108 Halle, Germany; hussainchem3@gmail.com; 4Department of Pharmacy Practice, Faculty of Pharmacy, King Abdulaziz University, P.O. Box 80260, Jeddah 21589, Saudi Arabia; aealtyar@kau.edu.sa; 5Pharmacy Program, Department of Pharmaceutical Sciences, Batterjee Medical College, P.O. Box 6231, Jeddah 21442, Saudi Arabia; mohamed.ashour@bmc.edu.sa; 6Department of Pharmacognosy, Faculty of Pharmacy, Ain Shams University, Cairo 11566, Egypt; 7University of Gastronomic Sciences, Piazza Vittorio Emanuele II 9, 12042 Pollenzo, Italy; a.pieroni@unisg.it; 8Department of Medical Analysis, Tiskh International University, Erbil 44001, Iraq

**Keywords:** ethnobotany, wild vegetables, Pathans, Gadoon Valley, Pakistan

## Abstract

**Simple Summary:**

To understand how traditional/folk biological knowledge changes across territories, cultures/languages, religions, and generations is crucial if we want to generate robust tools for preserving it. In this study we assessed the effect on foraging (gathering wild vegetables) of the affiliation to four different tribes within the same culture/language/religion in NW Pakistan. Through more than 100 interviews with local peoples conducted over a span of two years information about local wild vegetable names, growth habit, used plant parts, food/cooking details, medicinal perceptions, availability season, and market prices was collected. The survey recorded 51 non-cultivated vegetables while the dominant botanical families were Asteraceae and Fabaceae. Seven species were found to be sold at local and regional markets. Cross-cultural analysis among the wild plants foraged by the four considered tribes showed that the largest number of species was reported by members of the Hadarzai and Umarzai tribes, although most of the quoted wild vegetables were homogeneously gathered among all considered communities, with some more idiosyncratic plant uses among the Umarzai group, who have likely been less affected by the erosion of traditional knowledge or possibly have had less access to traded cultivated vegetables. This shows that food ethnobotanical knowledge exchanges overcome families and tribal boundaries, possibly through continuous social exchanges. The recorded food heritage will be essential for future projects aimed at fostering bio conservation, environmental sustainability, and food security.

**Abstract:**

The foraging and consumption of wild food plants is a long-standing tradition in many parts of the world and their importance in promoting food security has become more widely debated in recent years. The current study aimed to document, analyze, and interpret the traditional knowledge of non-cultivated vegetables among four Pathan tribes (Alisher Khel, Hadarzai, Haji Khel, and Umarzai) living in the Gadoon Valley, Swabi District, Khyber Pakhtunkhwa, NW Pakistan, and to evaluate how these practices vary among the considered tribal communities. A total of 104 informants were interviewed via a semi-structured, open-ended questionnaire and group discussions. The field survey was conducted from October 2018 to November 2020. Information about local names, growth habit, used plant parts, food/cooking details, medicinal perceptions, availability season, and market prices were collected. The field survey recorded 51 non-cultivated vegetables belonging to 24 botanical families, for which the frequently used plant parts included young leaves, stems, and flowers. The greatest number of use reports was recorded for *Colocasia* and the highest cultural index value was recorded for *Rumex dentatus*; the dominant botanical families were Asteraceae and Fabaceae (six species each). Seven species were found to be sold at local and regional markets. Cross-cultural analysis among the four considered tribes showed that the largest number of species was reported by members of the Hadarzai and Umarzai tribes, although most of the quoted wild vegetables were homogenously gathered among all considered communities, with some more idiosyncratic plant uses among the Umarzai group, who have likely been less affected by the erosion of traditional knowledge or possibly have had less access to traded cultivated vegetables. The novelty of the data was assessed by comparing it with the previously published wild food ethnobotanical literature of Pakistan, which showed fifteen new wild vegetables not yet reported in the NW of the country. The recorded food biocultural heritage should be seriously considered in future local development projects aimed at fostering environmental sustainability and food security.

## 1. Introduction

Human beings have always depended on plants for food and medicine. A large number of wild plants are used around the world for food in the form of fruits or vegetables. Wild food plants or wild edible plants refer to uncultivated plants that are used by local inhabitants as food [1,2]. In many developing countries, millions of people do not have sufficient food to meet their everyday dietary needs, and millions more are deficient in one or more micronutrients [3]. Wild vegetables can be valuable local crops that garner high prices in local and regional markets, contributing to the local cash income [4,5]. A literature review confirmed that wild vegetable uses have been documented in other parts of the country [6,7,8]: Ahmad et al. [9] reported 25 wild vegetables from seven divisions of Khyber Pakhtunkhwa, while Naveed and colleagues [10] investigated the ethnobotanical uses of 104 wild plants from Swabi District. The greatest number of species consumed as wild vegetables was recorded from the Chitral and Kurram districts [11,12]. In other parts of Asia, a remarkable diversity of gathered wild food plants has been generally documented in SE Asia, especially in the Lesser Himalaya region, Tibet, Vietnam, and inner China [13,14,15,16,17].

Pakistan is a developing country that ranks eleventh in the world in terms of food security risk [18]. Due to population size and natural and human-made disasters threatening local livelihood strategies and access to food, nearly 40% of households in northwest Pakistan are rated as food insecure [19,20]. A variety of wild plant species contribute to household food security and well-being [13]. Traditional knowledge concerning wild vegetables changes over time and space and is crucial for understanding patterns of evolution of folk nature and food knowledge. Socio-ecological knowledge and behaviors play an important role in the traditional use of wild vegetables. Wild vegetables play a part in the folk domestic provision of health and nutritional care in many parts of the world [9,21]; their role as traditional food-medicine has also been highlighted in some ethnobotanical studies [11,22,23,24]. Wild vegetables not only serve as alternatives to primary dietary items during food shortages, but also as a valuable food complement to everyday rural diets [25]; they can provide food to a large portion of rural populations but familiarity with wild vegetables is diminishing [26]. Despite the vast spread of agricultural activities, the trade of crops, and the large availability of cultivated vegetables all throughout the year, the practice of eating wild vegetables in several rural areas of the planet continues, due to their easy accessibility and their nutritional and health benefits [27]. In spite of the enormous increase in food production, 33% of rural inhabitants in developing countries are facing malnutrition and/or food shortages [23]. On the other hand, the world population is increasing at an alarming rate and is expected to increase to nine billion by the end of 2050, demanding 50–70% more food than we currently have available. More than 80% of the world’s population relies on just two dozen plant species for food [28]. Plant biodiversity is fundamental to addressing these challenges since at least 11% of the estimated 7000 edible plant species [29] are considered by the International Union for Conservation of Nature Red List as threatened. Foraging for wild food plants could become key to a more resilient, sustainable, biodiverse, and community participation-driven new “green revolution” [29], if knowledge of these natural resources can be properly “unlocked”. Additionally, some of these plants are still economically important in local small-scale markets and a source of income for locals, while some others have a specific ritual significance and are used on traditional and holy occasions [22].

The utilization of wild vegetables is an important local survival strategy during times of food shortage or drought [30]. However, the unsustainable utilization of some rare species or plant parts may lead to threats to the conservation of biodiversity [31]. Traditional knowledge of wild edible plants is, however, still partially neglected in ethnobotany [32] and has critically decreased due to ecological and socio-economic changes, as it is strongly tied to local heritage and, therefore, the cultural (linguistic, ethnic, religious) background of local communities [33].

Thus far, no ethnobotanical fieldwork has been conducted in the Gadoon Valley, nor within the specific context of locally gathered and consumed wild vegetables. The main objective of the current study is to analyze and document traditional knowledge regarding the utilization and marketing of wild vegetables among four Pathan tribes (Alisher Khel, Hadarzai, Haji Khel, and Umarzai) living in the Gadoon Valley.

The primary research objectives of the study were to:
(a)Explore and identify the wild vegetable resources collected in the Gadoon Valley;(b)Document their seasonality and eventual occurrence in local markets, as well as the local food knowledge attached to them and their possible medicinal perceptions;(c)Cross-culturally compare the recorded traditional knowledge among the four considered Pathan groups;(d)Compare the collected data with the published wild food ethnobotanical literature in order to identify possible novel reports of wild vegetable use.

## 2. Material and Methods

### 2.1. Study Area

Swabi District lies in the south and south-western part of Peshawar, Khyber Pakhtunkhwa, with an elevation ranging from 1181 to 7382 feet a.s.l. It lies between latitudes 34°15′39.4″ N and 72°41′04.6″ E (Figure 1). The north and north-eastern boundaries of the region are bounded by the Ambela (Buner) and Gadoon mountains. The Indus River forms the south and south-eastern border, while the west is bordered by the Nowshera and Mardan districts. The Gadoon region is hilly and occupies the north-eastern part of Swabi District. Of the total 27,441 ha, 8021 ha and 13,921 ha are occupied by agricultural lands and forests, respectively, while the remaining 5499 ha are rangelands [19].

### 2.2. Socio-Demographic Details of the Informants

Information about wild vegetables was collected through individual interviews and group discussions with the local population. A total of 104 informants from different cultural and educational backgrounds participated in the study (Table 1).

The Gadoon Valley is included in Swabi District, which is characterized by a large diversity of landscapes. The characteristics of the considered Pathan tribes are reported in Table 2. It is important to underline that members of the Hadarzai tribe reside in both mountain and plain areas, as this tribe is scattered all across the Gadoon Valley, having both urban and rural cultural properties.

### 2.3. Brief Historical Notes of the Gadoon Valley Tribes

The Gadoon area name derives from the Gadoon or Jadoon tribe living there. According to local accounts, this tribe arrived in the area during the 16th century with the purpose of crossing the Indus River and settling in the Hazara region. Two boats traversed the river, but the third party was diverted by the Utmanzai tribe, who prohibited their journey. They trace their descent to Ghurghusht (now Gadoon) and are named after their great-grandfather Muhammad Ashraf Ali Gadoon. The Gadoons are divided into two groups, Salars and Mansoors [34]. The Salars are further divided into the following subtribes: Alisher Khel, Haji Khel, Milli Khel, Shabi Khel, Mola Khel, Muhammad Khel, Yessa Khel, Bala Alisher Khel, Qalandar Khel, Sulaimanzai, and Khan Sher. The Salars residing in the Gadoon area are called Salarzai and are present in the villages of Mangal Chai, Gandaf, Chanai, and Dalorai. The people of the Hadarzai tribe are found in the following villages: Malakabad, Takail, and Qadra. Members of the Umarzai tribe reside in the villages of Sandwa, Gabasni, Bergali, Kund, Ganichatra, Utla, Amrai Bala, Amrai Payan, and Shengri.

There are about 85 villages in the Gadoon Valley, and they are divided into two areas. The lower plain area is called Hadarzai while the upper hilly and mountainous area is called Umarzai (Figure 2). All four tribes and their subtribes discussed above are found in the Gadoon area and often practice subsistence foraging and consume wild vegetables. Apart from these two groups, a third tribe (Hassazai) was given the right to use wasteland and forest called “Seri Khor”; they have a small population with a few families, but after some time they migrated to different areas of the country. The altitude of the area varies from 410 m a.s.l. on the eastern boundary of Mauza Gandaf to 2250 m a.s.l. at Shah Kot Sar (Mahaban Forest).

The four tribes are culturally, socially, and ritually slightly different from one another, although the majority of the customs are shared as they live together. Because of the numerous similarities in various aspects of daily life, the Hadarzai and Umarzai tribes maintain good relations and understanding and participate in engagements and wedding ceremonies together. The remaining tribes do not share that much in common, which makes each tribe unique and separate in many aspects. For example, each tribe has separate wedding and funeral traditions.

### 2.4. Overall Methodology and Data Collection

The overall methodological approach is reported in Figure 3.

Data about the traditional use of non-cultivated vegetables were collected in the Gadoon Valley from October 2019 to November 2020. The information was gathered through a semi-structured questionnaire, interviews, and focus-group discussions. A total of 104 key informants were selected, which included plant collectors, farmers, local sellers, housewives, and green grocers having thorough traditional knowledge of useful wild vegetables. All the interviews were conducted in Pashto, the local language of the communities. People were approached, specifically targeting elderly community members, in the mountains, at their homes, and in fields. Individuals were asked about their knowledge of non-cultivated vegetables (i.e., not only species considered botanically “wild”, but all vegetables that were not deliberately cultivated, possible including “escaped” cultigens gathered from the wild on a regular basis). All informants were locals and most of them lived in villages (Figure 4). The respondents were briefed about the aims and objectives of the study, and the prior informed consent of each participant was obtained; the fieldwork followed the Code of Ethics of the International Society of Ethnobiology (http://www.ethnobiology.net/what-wedo/core-programs/ise-ethics-program/code-of-ethics/code-in-english/, accessed on 13 June 2020). Older male participants facilitated the researchers in the collection of specimens, while younger individuals and older female respondents were more prone to share details about food preparations. It is important to note that we were not allowed to separately interview female community members in order to respect local customs, but they were interviewed in their homes by male family members, sharing their traditional knowledge about wild foods, which was then passed on to the researchers. We especially selected informants residing in the area and having continuous contact with the local environment, such as shepherds and elderly individuals, as they use wild vegetables on a daily basis. Questions about the consumption of wild edible vegetables focused on the local name of plant, part(s) of the plant used, seasonality and habitat, modalities of food processing, and ultimately medicinal perceptions and local market availability and prices.

### 2.5. Species Documentation and Identification

Voucher specimens of all documented plants were prepared. Identification was conducted using the available taxonomic literature linked to the flora of Pakistan [35,36]. The vouchers were subsequently deposited in the Herbarium of the Department of Botany at the University of Peshawar.

### 2.6. Data Analysis

#### 2.6.1. Jaccard Similarity Index

Data collected among the four tribes were compared via Venn diagrams and by calculating the Jaccard (similarity) index (JI) for each pair of datasets. To determine the similarity between the two sets of data, the following formula was used:J I = (the number in both sets)/(the number in either set) ∗ 100

The formula in notation is as follows:J (X, Y) = |X ∩ Y|/|X ∪Y|

Furthermore, the documented data were also compared with the wild vegetable ethnobotanical literature of Pakistan in order to assess their possible novelty.

#### 2.6.2. Use Report and Cultural Importance Indexes

The cultural importance of each species in every tribe was calculated by analyzing the cultural importance index (CI). It is calculated using the formula:CI = URi/Ni
where UR is the use report in each tribe for each taxon and Ni is the number of informants in every tribe.

The mean cultural importance index (mCI) for each species was calculated according to the quantitative ethnobiological literature [13,37,38,39].

## 3. Results and Discussion

### 3.1. Wild Vegetables in the Gadoon Valley

The survey recorded 51 wild vegetables belonging to 24 families consumed by the four tribal communities of the Gadoon Valley, Swabi District. The study represents the first attempt to report the local names of wild folk vegetables among the considered Pathan communities. For each folk taxon, we reported the local name, part(s) used, growth habit, availability period, medicinal properties, and mode of preparation (Table 3). *Asphodelus tenuifolius*, *Berberis lycium*, *Chenopodium murale*, *Ficus palmata*, *Mentha longifolia*, *Rumex dentatus*, *Solanum nigrum*, *Tulipa stellata*, and *Zanthoxylum armatum* were the most commonly used species in the area and consumed as vegetables by all the tribal groups.

The dominant families were Asteraceae and Fabaceae (six species each), followed by Brassicaceae (five species); Amaranthaceae and Polygonaceae (four species each); and Plantaginaceae, Malvaceae, Lamiaceae, Caryophyllaceae, Apiaceae, and Alliaceae (two species each). Most of the wild vegetables included in families such as the Asteraceae, Fabaceae, Brassicaceae, and Malvaceae are widespread and have been documented as wild plant ingredients in other countries of Eurasia [40,41,42,43]. The most represented families were Polygonaceae, Lamiaceae, Apiaceae, Amaryllidaceae, Asteraceae, and Malvaceae [44]. The prevalence of fruits and vegetables in wild plant foods has been highlighted [45,46,47].

Most of the wild taxa were herbs (48 species), but there were also two medium-sized trees, namely, *Bauhinia variegata* and *Zanthoxylum armatum*, and one shrub, *Berberis lycium.* Generally, all the wild vegetables, except species that were restricted to a particular location, such as *Bidens pilosa*, were present everywhere throughout the area: in crop fields, on the banks of rivers and streams, on hilly terrain, and on the lower slopes of hills. Shepherds were the most knowledgeable informants, demonstrating the significance of the relationship between wild vegetable resilience and the survival of pastoralism. However, the transmission of ethnobotanical knowledge from elderly individuals to the younger generation is continuously decreasing due to the generation gap and fast-changing lifestyles.

The plant parts consumed included leaves, fruit, young shoots, and stems. Leaves were the most used plant part, especially in salads, as raw snacks, and as cooked vegetables, but in case of *Bauhinia variegata* the flowers were used while *Berberis lycium* roots were used commercially. For some species, such as *Solanum nigrum*, *Caralluma tuberculate*, and *Allium griffithianum*, the whole plant was used in salads and as cooked vegetables. Species such as *Commelina benghalensis*, *Silene conoidea*, and *Trifolium repens*, gathered from crop fields, represented wild weeds. Wild vegetables were consumed in the early stages of life and mostly the aerial parts and green leaves are used for cooking. Some species like *Oxalis corniculata* and *Rumex vesicarius* were consumed raw. Usually, green leaves were used, but the floral buds of *Bauhinia variegata* and the young fruit of *Ficus palmata* were cooked. The leaves of some species, including that of *Mentha longifolia*, *Mentha spicata*, and *Bauhinia variegata*, were stored dried for cooking.

### 3.2. Seasonality and Market Value of Wild Vegetables

The availability of wild vegetables largely depends on the climate of an area. The largest number of species is available from March to October (10 species), while availability decreases in November and reaches a minimum from December to February. The duration of availability varies from two to eight months (Table 4).

In the study we also observed that some species are sold in local markets of the study area. The leaves of *Colocasia* spp., *Mentha longifolia*, *Mentha spicata*, and *Rumex dentatus* are routinely sold in local markets during the available season. Prices vary on the basis of demand, supply, and season of wild vegetables. The current study also reported market prices for the first time; e.g., *Zanthoxylum armatum* (USD 2.50), *Berberis lycium* (USD 1.88), and *Bauhinia variegata* (USD 0.94) (Figure 5). Dried flowers of *Bauhinia variegata* and the roots of *Berberis lycium* are sold and consumed in large quantities (Figure 6). *Zanthoxylum armatum* is commercially sold and used for seasoning. Although a quantitative analysis of the commercial impact of foraging was not the main objective of the current study, our investigation suggests that the small-scale market of these wild vegetables can generate income, which may be crucial in disadvantaged households.

### 3.3. Wild Vegetables in Local Folk Cuisine and Domestic Medicine

The simple cooking method used for most of the wild vegetables was as follows: onions are fried in oil or ghee together with some condiments (tomatoes, garlic, green chilies, coriander, turmeric, or mint), depending on their availability and taste preferences. Soft leaves of certain vegetables, such as *Solanum nigrum*, *Colocasia* spp., *Rumex* spp., and *Brassica carinata*, are chopped into pieces and directly cooked (Figure 7). In contrast, a few vegetables, namely, *Bauhinia variegata*, *Chenopodium album*, and *Nasturtium officinale*, go through an interesting culinary process: they are collected and sun-dried, then the plant parts are ground, fried in oil, and, after the addition of milk, boiled for two hours until it forms a thick, viscous porridge-like soup (Figure 8). Young stems of *Asphodelus tenuifolius* are similarly chopped, mixed with bread, and then cooked.

Alternatively, some plants, such as *Oxalis corniculate, Rumex vesicarius*, and *Vicia sativa*, are consumed raw, both in salads and (very few species) as a snack. The persistence of very few raw snacks in the study area can be linked to the decline of pastoralist activities, as herding has been found to often shape the custom of eating, on the spot, edible plants found in pastures [49,50,51,52], and herders still often eat more raw snacks than local horticulturists [49].

Some of the quoted wild vegetables were also used as folk medicines. The current field study revealed that a large portion of the quoted wild vegetables were perceived to have a medicinal value. Thirty-eight species of wild vegetables were quoted as used for treating or mitigating different ailments. For example, the young leaves of *Amaranthus spinosus* and *Amaranthus viridis* were used for treating diarrhea. In addition, the leaves and fruits of *Berberis lycium* were used to treat diabetes, while decoctions of its roots were used to treat bone fractures and back pain. The young stems of *Asparagus officinalis* were used to treat constipation, *Bistorta amplexicaulis* and *Medicago denticulata* were used to treat dysentery, and the chopped leaves of *Oxalis corniculate* were considered effective as an emetic remedy. Members of the Umarzai and Haji Khel communities seemed to possess less knowledge of the possible medicinal applications of wild vegetables, which cannot be linked to a more limited availability of medicinal wild vegetables, as the Umarzai tribe lives in more isolated mountain areas, but rather to a limited understanding and/or exposure to medicinal information and representations of plants coming from urban mass-media.

Fruits and vegetables have received increased attention in promoting health due to the protective properties of the non-bioactive compounds they contain, which increases their use in human diets [53]. The green leaves of *Mentha longifolia*, *Mentha spicata*, and *Oxalis corniculata* are used for alleviating diarrhea, reducing fever (antipyretic), and treating stomach disorders, respectively [54,55,56]. Leaves are the preferred plant part for both vegetables and medicines, while rhizomes, seeds, and aerial parts may also be used therapeutically. The dual use of species as food and medicine in traditional societies indicate a continuum between medicinal and edible plants and may reflect their shared origin and interconnection [57]. *Convolvulus arvensis*, *Eclipta prostrata*, *Malva neglecta*, *Nasturtium officinale*, and *Chenopodium album* are used to treat urinary tract infections, constipation, and diarrhea, and to remove dandruff. Some of the wild vegetables observed in this study have limited medicinal usage because of their sour taste, i.e., *Commelina benghalensis*, *Ficus palmata*, *Lactuca serriola*, and *Lathyrus aphaca.*

### 3.4. Quantitative Analysis

A use report is defined as the number of respondents that mention a particular species during interviewing [58,59,60,61]. The use report is applied in determining the plant with greatest number of uses (most frequently used) for the treatment of an ailment or disease. Informants may mention several medicinal uses for one species. The largest number of use reports was found for *Colocasia* spp. (97), followed by *Bauhinia variegata* (93), *Mentha spicata* (90), *Mentha longifolia* (81), and *Berberis lycium* (79). The wild vegetables having the fewest use reports were *Trifolium repens* and *Pimpinella saxifraga* (5 each).

To quantify the importance of wild vegetables, the cultural importance index (CI) and mean CI (mCI) were calculated. The cultural index mainly depends on the range, value, and uses of a species in a given area. The cultural importance index explains not only the spread of uses (number of informants) for each species, but also its versatility, i.e., the diversity of its uses [62,63,64]. The highest mCI values were recorded for *Rumex dentatus* (0.898), *Bauhinia variegata* (0.894), *Mentha spicata* (0.865), *Mentha longifolia* (0.79), *Berberis lycium* (0.76), *Zanthoxylum armatum* (0.721), and *Solanum nigrum* and *Amaranthus viridis* (0.702 each) (Table 3). The plant having the lowest mCI value was *Tulipa stellata* (0.142).

The family cultural importance index (fCI) was also calculated for each plant family (Figure 9). The highest recorded value of fCI was for Amaranthaceae (2.133), followed by Polygonaceae (1.725), Asteraceae (1.689), Brassicaceae (1.165), Lamiaceae (1.655), and Fabaceae (1.337); the family having the lowest mCI value was that of Apocynaceae (0.116).

### 3.5. Cross-Cultural Comparison among the Four Considered Pathan Tribes

Of the 51 total wild vegetables, only nine species were gathered and consumed by all four tribes (*Asphodelus tenuifolius*, *Chenopodium murale*, *Berberis lycium*, *Ficus palmata*, *Rumex dentatus*, *Solanum nigrum*, *Mentha longifolia*, *Tulipa stellata*, and *Zanthoxylum armatum*).

Cross-cultural comparison shows that wild vegetable use among the four studied tribes of the Gadoon Valley is quite similar. The wild vegetables used by the studied communities and their related Jaccard similarity indexes are indicated in the Venn diagram presented in Figure 10. Both ecological and social factors may have played a role in shaping commonalities and differences among the studied communities. The Alisher Khel group, for example, seems to show a restricted use of wild edible plants compared to the other communities, possibly because of limited availability of certain plant species in the plain area and cultural adaptation to a modern lifestyle (erosion of traditional knowledge); they in fact most use *Chenopodium album* and *Portulaca oleracea*, which are ignored by other communities. The Umarzai tribe (living in a mountainous environment) showed the greatest number (eight) of idiosyncratic plant uses (i.e., uses not shared with the other groups), which can be only partially linked to a richer mountain plant biodiversity, as only one species, *Bistorta amplexicaulis*, clearly especially grows in mountain pastures. The distinctive plant uses of this group are much more likely due also to isolation, i.e., less erosion of traditional knowledge and/or less access to traded cultivated vegetables. However, the largest number of species was reported by members of both the Hadarzai and Umarzai tribes, which live in very different landscapes and have very different predominant occupations. Moreover, high similarity indexes among the different groups reveal that traditional ecological knowledge has, in fact, followed the path of homogenization, perhaps due in part to the fact that (apart from the Hadarzai groups) intermarriages are common. Most significantly, the Hadarzai did not show the use of many idiosyncratic plants (only two genera), even though they are mainly endogamous, in contrast to the other three groups. This may be due to the fact that they live scattered across the entire valley and therefore transmission of traditional plant knowledge have occurred not only within kinship networks, but possibly also via social exchanges, which may have generated osmotic effects among the diverse groups.

Differences in plant use among the different groups appear to be quite moderate (Figure 11).

## 4. Novelty of the Recorded Data and Future Perspectives

The published literature on wild vegetables has demonstrated that wild food plant foraging is still very much alive and robust in rural NW Pakistan [9,10,11,12,24,59]. The comparative analysis we conducted with these pre-existing ethnobotanical studies revealed that fifteen wild edible vegetables have not been previously documented from Khyber Pakhtunkhwa: *Allium jacquemontii*, *Asphodelus tenuifolius*, *Bidens pilosa*, *Brassica carinata*, *Colocasia* spp., *Eclipta prostrata*, *Lactuca serriola*, *Medicago denticulata*, *Pimpinella saxifraga*, *Silene vulgaris*, *Sisymbrium orientale*, *Solanum nigrum*, *Tulipa stellata*, *Veronica polita*, and *Zanthoxylum armatum.*

Among these fifteen taxa, seven species were also sold in local markets. While some of these species grow in mountain environments, half of them (*Brassica*, *Lactuca*, *Medicago*, *Silene*, *Sisymbrium*, *Solanum*, and *Veronica* spp.) are represented by synanthropic species, i.e., plants growing in highly anthropogenic environments. The fact that these taxa, already recognized as being consumed in other areas of Pakistan, have been rarely recorded in the NW of the country may be due to the fact that these plants are normally gathered by women in the vicinity of houses and therefore these practices tend to be “more invisible” to researchers than those related to higher mountain plants normally gathered by male community members. Moreover, these wild vegetables are perceived—again, often by women who prepare the food and are the domestic care givers—as having high nutritional value in daily cuisine. These newly reported wild vegetables are also medicinally perceived as well, since several of them are used as a diuretic and a liver tonic, and for treating ulcers, diarrhea, and menstrual disorders.

It is interesting to note that these synanthropic species are very commonly used in Near Eastern and Mediterranean cuisine [40,41]; however, a direct historical link between the investigated Pathan groups and the Near East could not be detected. Moreover, Mediterranean and Near Eastern modalities of preparing/cooking synanthropic vegetables are quite different as they are often enjoyed in salads or fried in olive oil and garlic, whereas in the study area these wild ingredients are often consumed raw or cooked with onions, ghee, and spices.

In our study, we also found that some wild vegetables were sold in the area, and we reported the prices for seven of these plants. This capacity of generating income for families and activating small-scale economies could also be crucial for the resilience of these foraging practices in the near future. Moreover, wild plant species play an important role in the nutritional health of rural communities by providing cheap sources of nutrients, and thus the possible nutraceutical properties of wild vegetables should be promoted for implementing a more community-centered public health approach. In mountainous areas of Pakistan, as in other developing regions, livelihoods are mainly based on subsistence horticulture, animal farming, and the communal use of pastures and forests. Some communities in hilly and mountainous areas are highly dependent on wild food products, due to their low income and inaccessibility to “urban” food products. The current research can provide baseline data for communities aiming to implement food security and sovereignty by mitigating the erosion of local knowledge associated with these ingredients. More field research in other NW Pakistani “marginal” regions is needed, however, in order to better understand the complex interaction between livestock activities, traditional food heritage, and rural landscapes.

More importantly, this study clearly shows a remarkable homogeneity of plant knowledge among the four considered tribal groups (with the partial exception of one group), perhaps indicating that the social exchange of knowledge and practices outside the household, and even outside the clans/tribes, has been crucial in homogenizing possible pre-existing distinctive food traits of each group.

Moreover, according to our study participants, wild food plants availability decreased significantly in certain contexts as a result of increasing anthropogenic pressures, such as unhealthy farming environments, the availability of cultivated vegetables, the penetration of industrial food, environmental change and degradation, cultural changes in the gendered division of labor, as well as the weak general governance of food policies and economies at both the regional and national level, resulting in a substantial loss of ecosystem services [64]. Additionally, the loss of traditional knowledge is inexorable and significant among the younger generations. It is therefore essential to help individuals, especially those of younger generations, re-discover wild plant sources for food and small-scale commercial purposes, as well as for improving their overall well-being and as a response to the alarming growth of the human population and decrease of income that can be observed in many areas of Pakistan and the world [8,33].

## 5. Conclusions

Our study reported 51 wild vegetables gathered and consumed as part of the local bio-cultural heritage among four Pathan groups in the Gadoon Valley. The cross-cultural comparison revealed a high homogeneity of plant uses, with a remarkable predominance of wild vegetables gathered among the Umarzai and Hadarzai tribes; the former group also retained a few vegetables which were not reported by the other groups, possibly as a result of their mountain environment and stronger isolation, presumably indicating both minor erosion of traditional knowledge and more restricted access to traded cultivated vegetables. However, the highly homogenous foraging practices among the study groups suggest that the social exchange (i.e., horizontal transmission) of plant knowledge may be crucial for sharing foraging practices and their associated cuisines, especially if the considered cultural groups share the same religion and commonly intermarry. While in other studies we have clearly demonstrated that religious and sometimes linguistic affiliations represent important factors shaping food customs and the domestic uses of foraged ingredients [11,42,64], the current study seems to suggest that these divides are blended among tribes of the same ethnic/religious group.

Further studies are needed at the interface between biological and cultural diversity to more systematically assess variations and commonalities in plant knowledge among diverse ethnic, linguistic, and religious groups, and to evaluate the variability within them. This could provide some important elements for further understanding how this knowledge of nature is transmitted and how it changes and evolves [65].

Future in-depth studies will be needed to investigate the impacts of global and local changes on diverse cultural practices and beliefs relating to traditional gastronomy. Producing small-scale food products requires knowledge and special skills that need to be constantly maintained and updated. The local people of the Gadoon Valley collect these wild vegetables in different seasons and their food elaboration is routinely done by women. The findings of this study could serve as a baseline for promoting eco-tourism and supporting sustainable development initiatives. A few of the recorded wild food plants are sold in local markets (e.g., *Colocasia*, *Mentha*, *Rumex*, *Bauhinia*, *Berberis*, and *Zanthoxylum* spp.), which could be useful for local wild food plant-centered projects aimed at revitalizing traditional ecological knowledge and generating small-scale economies.

## Figures and Tables

**Figure 1 biology-10-00537-f001:**
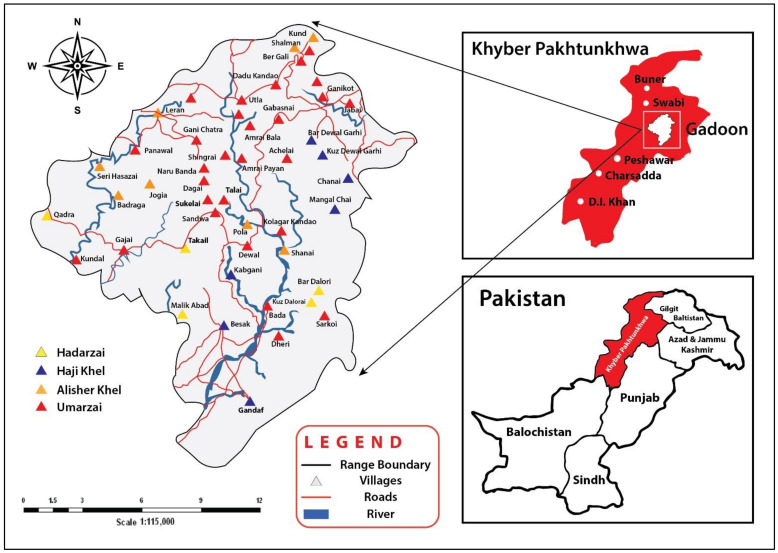
Map of the study area and visited villages.

**Figure 2 biology-10-00537-f002:**
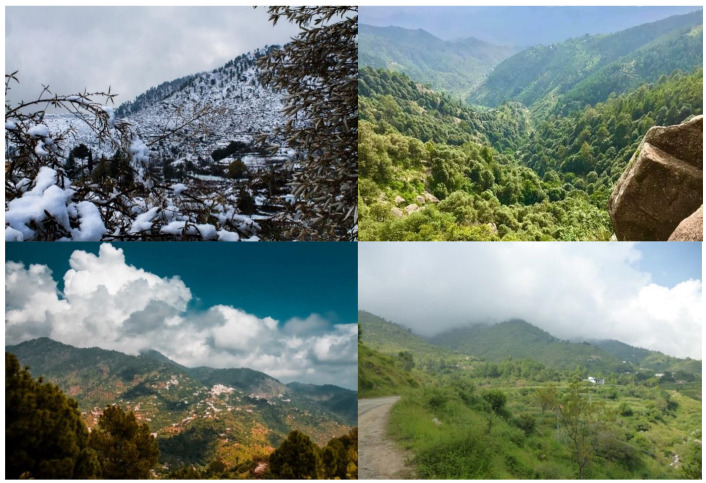
Landscapes of the study area in different seasons.

**Figure 3 biology-10-00537-f003:**
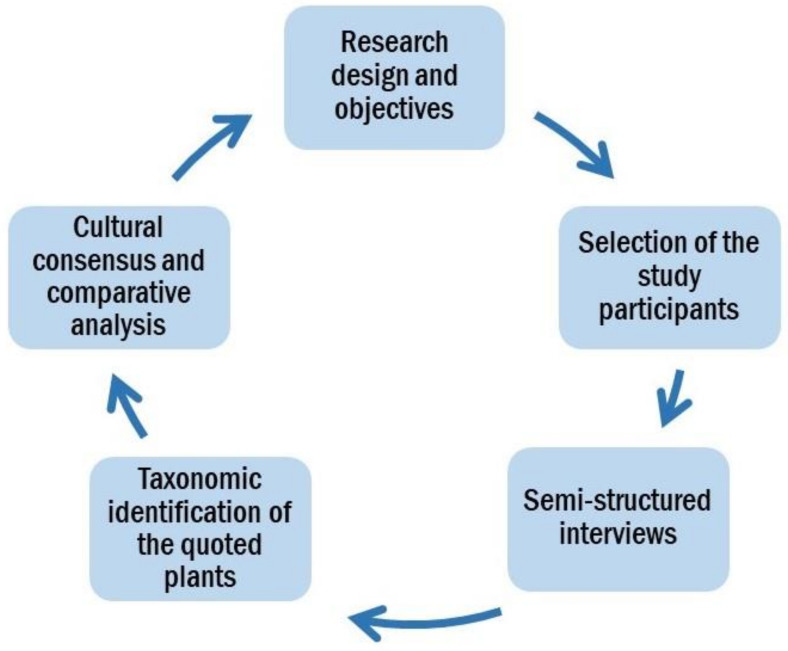
Overall methodological scheme.

**Figure 4 biology-10-00537-f004:**
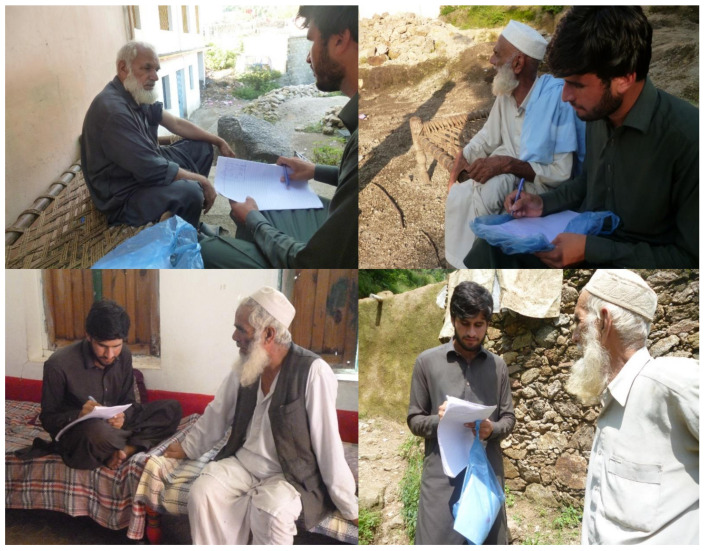
Interviewing local elderly study participants during the fieldwork.

**Figure 5 biology-10-00537-f005:**
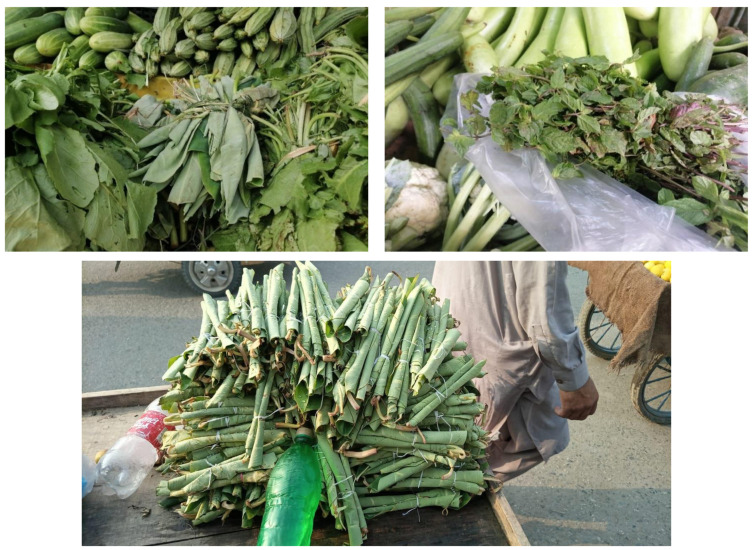
*Colocasia* and *Mentha spicata* for sale at the local market.

**Figure 6 biology-10-00537-f006:**
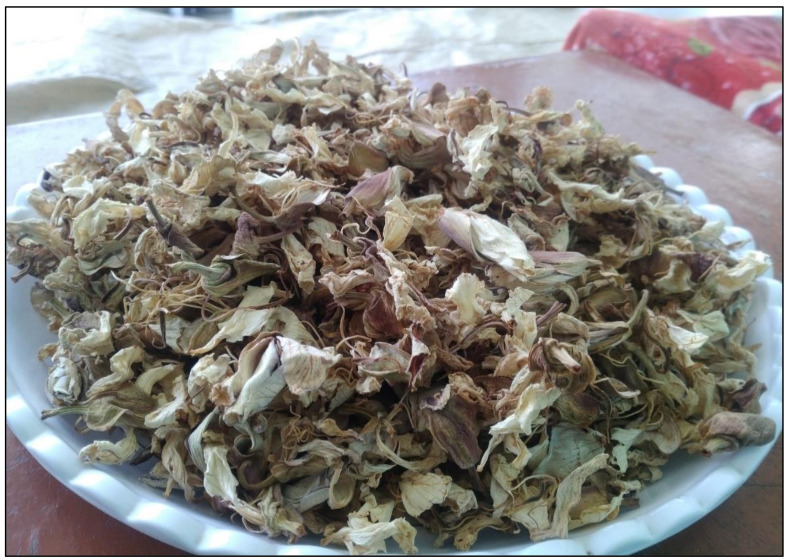
Dried flowers of *Bauhinia variegata*.

**Figure 7 biology-10-00537-f007:**
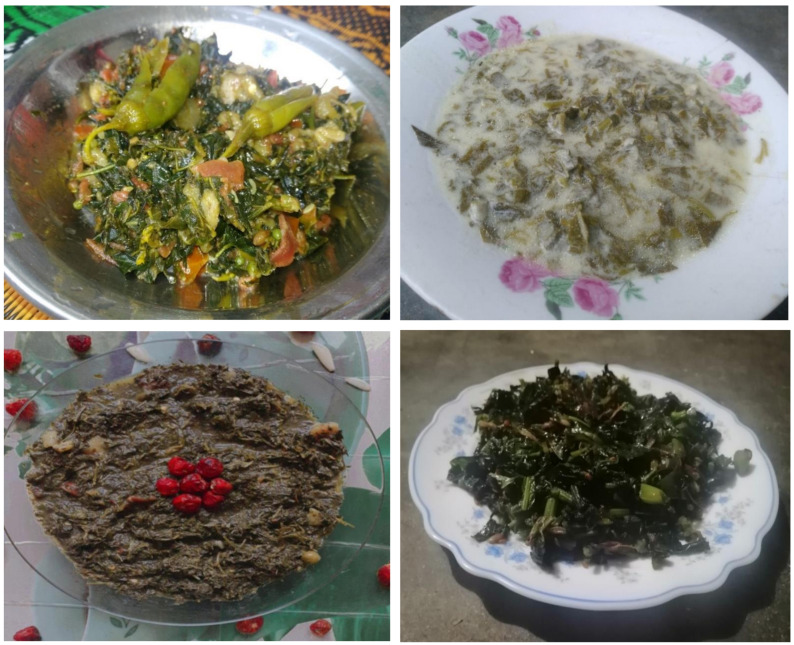
Some wild edible vegetable-based dishes cooked in the homes of the study participants.

**Figure 8 biology-10-00537-f008:**
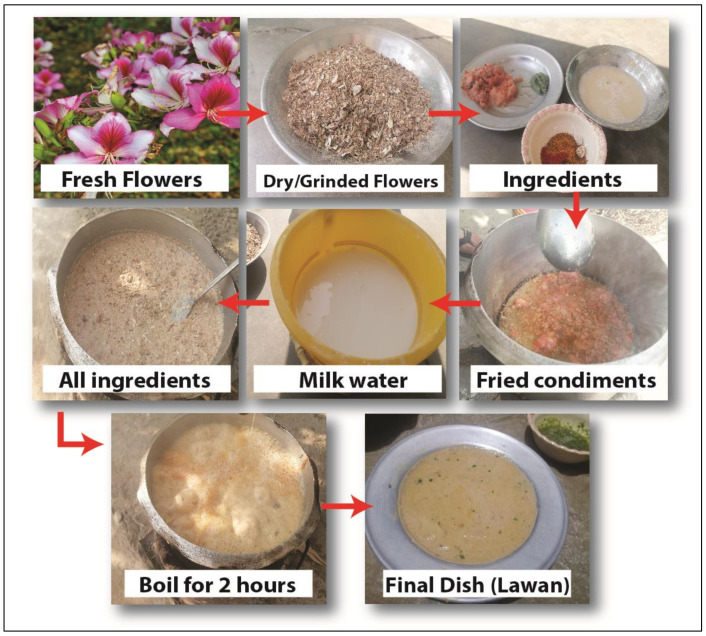
The various steps in the preparation of Lawan.

**Figure 9 biology-10-00537-f009:**
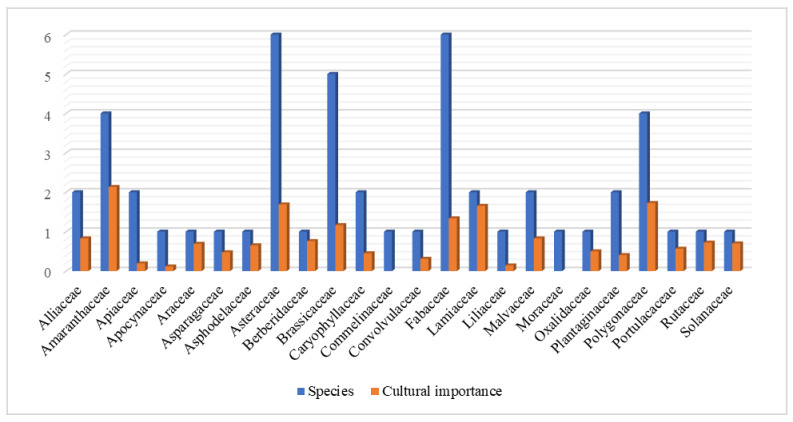
Cultural importance of quoted botanical families.

**Figure 10 biology-10-00537-f010:**
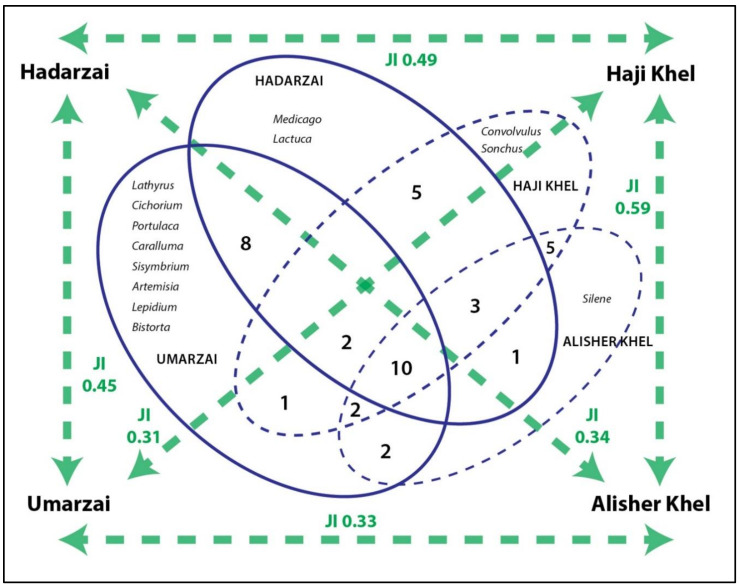
Venn diagram showing the overlap of wild vegetables (genera) use among the four considered groups, as well as the Jaccard similarity indexes.

**Figure 11 biology-10-00537-f011:**
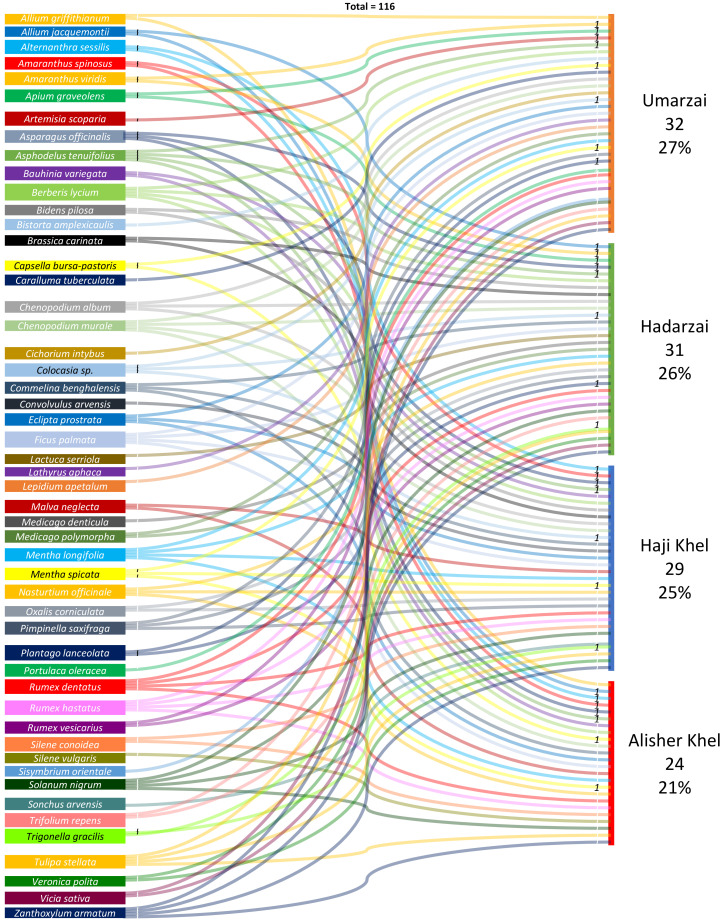
Alluvial diagram illustrating the distribution of plant uses among the four considered Pathan tribes.

**Table 1 biology-10-00537-t001:** Demographic characteristics of the study participants.

Parameters	Classes	Frequency of Respondents in Each Class	Percentage
Gender	Male	88	84.62
	Female	16	15.38
Age range	20–40 years	19	18.26
	40–60 years	31	29.90
	above 60 years	54	51.93
Educational background	Illiterate	47	45.19
	Elementary	29	27.88
	Intermediate	17	16.35
	Graduate	9	8.66
	Post-graduate	2	1.92
Occupation	Plant collectors	39	37.5
	Farmers	29	27.89
	Local sellers	12	11.54
	Housewives	16	15.38
	Shopkeepers	8	7.69

**Table 2 biology-10-00537-t002:** Main ecological and social characteristics of the study groups.

Tribe	Number of Villages	Average Elevation (Feet) and Ecology	Intermarriages	Population	Main Occupation	Interviewed Study Participants	
Umarzai	30	5134.5, mountainous area	Intermarriages with Haji Khel and Alisher Khel	47,025	Farmers, shepherds	23 males	3 females
Hadarzai	17	1486.2, both mountain and plain areas	Marriages within the tribe only	31,900	Shopkeepers	22 males	4 females
Haji Khel	23	1561.7, plain area	Intermarriages with Umarzai and Alisher Khel	46,015	Laborers, local sellers	21 males	5 females
Alisher Khel	15	3477.7, both mountain and plain areas	Intermarriages with Haji Khel	15,296	Collectors, farmers	22 males	4 females

**Table 3 biology-10-00537-t003:** Non-cultigens gathered and consumed in the Gadoon Valley.

Species Number	Plant Species or Taxon, Botanical Family, and Voucher Code	Local Name	Used by Tribal Groups				Parts Used	Habit	Price/kg in Local Markets (USD)	Perceived Medicinal Properties	Local Food Preparation	UR	mCI	Food Uses Previously Reported in KP, Pakistan
			UZ	HZ	HK	AK								
1.	*Allium griffithianum* Boiss.; Alliaceae; B.She.150.UOP	Zangali pyaz	+	-	-	+	Whole Plant	Herb	-	None	Cooked, seasoning	-	-	[11]
2.	*Allium jacquemontii* Kunth; Alliaceae; B.She.151.UOP	Sor pyaz	-	+	-	+	Leaves	Herb	-	None	Cooked, seasoning	-	-	No
3.	*Alternanthera sessilis* (L.) R.Br. ex DC.; Malvaceae; B.She.152.UOP	Soba	-	-	+	+	Leaves, shoots	Herb	-	Hepatitis, hair tonic	Cooked	30	0.385	[10]
4.	*Amaranthus spinosus* L.; Amaranthaceae; B.She.153.UOP	Ganhar	-	-	+	+	Leaves	Herb	-	Diarrhea, wounds	Cooked	70	0.673	[9,24]
5.	*Amaranthus viridis* L.; Amaranthaceae; B.She.154.UOP	Senglai	+	+	-	-	Leaves	Herb	-	Laxative	Cooked	73	0.702	[9,10,11,48]
6.	*Apium graveolens* L.; Apiaceae; B.She.155.UOP	Danai	+	+	-	-	Leaves and shoots	Herb	-	None	Salad, seasoning, and cooked	-	-	[24,48]
7.	*Artemisia scoparia* Waldst & Kitam.; Asteraceae; B.She.156.UOP	Jukay	+	-	-	-	Stem, leaves and shoots	Herb	-	Jaundice and hepatitis	Salad	19	0.244	[10]
8.	*Asparagus officinalis* L.; Asparagaceae; B.She.157.UOP	Saboon botai	-	+	+	+	Stem	Herb	-	Constipation	Cooked	37	0.475	[11]
9.	*Asphodelus tenuifolius* Cav.; Asphodelaceae; B.She.158.UOP	Ogakai	+	+	+	+	Leaves	Herb	-	Diuretic, ulcers	Cooked	51	0.654	No
10.	*Bauhinia variegata* L.; Fabaceae; B.She.159.UOP	Kulyar	-	-	+	+	Flowers	Tree	$0.94	Thyroid hormone-regulating activity	Cooked	93	0.894	[9]
11.	*Berberis lycium* Royle.; Berberidaceae; B.She.160.UOP	Karoskai	+	+	+	+	Leaves and fruit, shoots	Shrub	$1.88	Diabetes, bone fractures	Cooked	79	0.76	[12]
12.	*Bidens pilosa* L.; Asteraceae; B.She.161.UOP	Sormal	-	+	+	-	Leaves	Herb	-	Leprosy and skin cuts	Cooked	7	0.27	No
13.	*Bistorta amplexicaulis* (D. Don) Greene; Polygonaceae; B.She.162.UOP	Gule rana	+	-	-	-	Leaves and roots	Herb	-	Dysentery	Cooked	12	0.231	[11]
14.	*Brassica carinata* (Braun) O.E Schulz; Brassicaceae; B.She.163.UOP	Ghat, sharsham	-	+	+	-	Leaves	Herb	-	Menstrual disorder	Cooked	6	0.231	No
15.	*Capsella bursa-pastoris* (L.). Medik.; Brassicaceae; B.She.164.UOP	Badshah	+	-	-	+	Leaves	Herb	-	Diarrhea, bladder infections	Salad, cooked	21	0.270	[48]
16.	*Caralluma tuberculata* N.E.Br.; Apocynaceae; B.She.165.UOP	Pamankai	+	-	-	-	Whole plant	Herb	-	Diabetes mellitus	Salad	9	0.116	[9,11,24,48]
17.	*Chenopodium album* L.; Amaranthaceae; B.She.166.UOP	Sarmai	+	+	+	-	Shoots and leaves	Herb	-	Urinary tract infections	Cooked	40	0.385	[9,10,11,12,48]
18.	*Chenopodium murale* L.; Amaranthaceae; B.She.167.UOP	Thor sarmai	+	+	+	+	Leaves	Herb	-	Anthelmintic	Cooked	29	0.373	[9,10]
19.	*Cichorium intybus* L.; Asteraceae; B.She.168.UOP	Shinkai	+	-	-	-	Leaves	Herb	-	Gastrointestinal ailments	Cooked, salad	31	0.596	[9]
20.	*Colocasia* sp.; Araceae; B.She.169.UOP	Narai kachaloo	+	+	+	-	Leaves and fruit	Herb	$0.63	Diarrhea	Cooked and salad	97	0.692	No
21.	*Commelina benghalensis* L.; Commelinaceae; B.She.170.UOP	Nari	-	+	+	+	Leaves	Herb	-	None	Cooked	-	-	[11]
22.	*Convolvulus arvensis* L.; Convolvulaceae; B.She.171.UOP	Pervati	-	-	+	-	Leaves and shoots	Herb	-	Constipation and remove dandruff	Cooked	13	0.311	[11]
23.	*Eclipta prostrata* (L.) L.; Asteraceae; B.She.172.UOP	Bandakai	+	-	+	+	Leaves and shoots	Herb	-	Liver tonic	Cooked	27	0.347	No
24.	*Ficus palmata* Forsskal.; Moraceae; B.She.173.UOP	Enzar	+	+	+	+	Leaves and shoots	Tree	-	None	Cooked	-	-	[24]
25.	*Lactuca serriola* L.; Asteraceae; B.She.174.UOP	Kokara	-	+	-	-	Leaves	Herb	-	None	Cooked	-	-	No
26.	*Lathyrus aphaca* L.; Fabaceae; B.She.175.UOP	Zyar mattar	+	-	-	-	Leaves and fruit	Herb	-	None	Cooked	-	-	[9.11]
27.	*Lepidium apetalum* Willd.; Brassicaceae; B.She.176.UOP	Bashkai	+	-	-	-	Leaves	Herb	-	None	Cooked	-	-	[9]
28.	*Malva neglecta* Wallr.; Malvaceae; B.She.177.UOP	Panerak	-	-	+	+	Leaves	Herb	-	Urinary tract infections	Cooked, salad	46	0.442	[11]
29.	*Medicago denticulata* Willd.; Fabaceae; B.She.178.UOP	Lewani	-	+	-	-	Leaves	Herb	-	Dysentery	Cooked	13	0.250	No
30.	*Medicago polymorpha* L.; Fabaceae; B.She.179.UOP	Shpaishtai	+	+	-	-	Leaves	Herb	-	None	Cooked	-	-	[9,10,11]
31.	*Mentha longifolia* (L.) L.; Lamiaceae; B.She.180.UOP	Velanai	+	+	+	+	Leaves	Herb	$0.31	Diarrhea	Salad, cooked	81	0.79	[9,10,11,12,48]
32.	*Mentha spicata* L.; Lamiaceae; B.She.181.UOP	Podina	+	-	+	+	Leaves	Herb	$0.44	Stomach ache, intestinal pains	Salad	90	0.865	[11,48]
33.	*Nasturtium officinale* R.Br.; Brassicaceae; B.She.182.UOP	Tarabera	-	+	+	+	Leaves and stem	Herb	-	Urinary tract infections	Cooked, salad	69	0.664	[9,10,11,12]
34.	*Oxalis corniculata* L.; Oxalidaceae; B.She.183.UOP	Tarokai	-	+	+	-	Leaves, stem	Herb	-	Vomiting	Salad	52	0.500	[10,11,24,48]
35.	*Pimpinella saxifraga* L.; Apiaceae; B.She.184.UOP	Ogai	+	+	-	-	Leaves	Herb	-	Indigestion	Cooked	5	0.193	No
36.	*Plantago lanceolata* L.; Plantaginaceae; B.She.185.UOP	Isphagol	+	+	-	-	Stem, leaves	Herb	-	Skin irritations	Cooked	42	0.404	[11]
37.	*Portulaca oleracea* L.; Portulacaceae; B.She.186.UOP	Orkharai	+	-	-	-	Shoots	Herb	-	Antiseptic	Cooked	59	0.567	[11]
38.	*Rumex dentatus* L.; Polygonaceae; B.She.187.UOP	Shalkhai	+	+	+	+	Leaves	Herb	$0.63	Kidney stones	Cooked	70	0.898	[9,10,11,12]
39.	*Rumex hastatus* D. Don.; Polygonaceae; B.She.188.UOP	Narai shalkhai	+	+	+	+	Leaves	Herb	-	Kidney stones	Cooked	37	0.596	[9,12]
40.	*Rumex vesicarius* L.; Polygonaceae; B.She.189.UOP	Ghat tarokai	+	+	-	-	Leaves	Herb	-	None	Salad	-	-	[48]
41.	*Silene conoidea* L.; Caryophyllaceae; B.She.190.UOP	Mangotai	-	-	+	+	Fruit and leaves	Herb	-	Anemia	Cooked	13	0.168	[9,10,11]
42.	*Silene vulgaris* (Moench) Garcke; Caryophyllaceae; B.She.191.UOP	Sor mangotai	-	-	-	+	Leaves and shoots	Herb	-	Constipation	Salad, cooked	21	0.283	No
43.	*Sisymbrium orientale* L.; Brassicaceae; B.She.192.UOP	Orai	+	-	-	-	Leaves	Herb	-	None	Eaten raw as a snack	-	-	No
44.	*Solanum nigrum* L.; Solanaceae; B.She.193.UOP	Kachmacho	+	+	+	+	Whole	Herb	-	Laxative, appetite stimulant	Cooked	73	0.702	No
45.	*Sonchus arvensis* L.; Asteraceae; B.She.194.UOP	Zyar gulai	-	-	+	-	Leaves	Herb	-	Asthma, chest pain	Boiled or variously cooked	18	0.232	[48]
46.	*Trifolium repens* L.; Fabaceae; B.She.195.UOP	Shutal	+	+	-	-	Leaves	Herb	-	Cough, fever	Cooked, salad	5	0.193	[11]
47.	*Trigonella gracilis* Benth.; Fabaceae; B.She.196.UOP	Zyar shpaishtai	-	+	+	-	Leaves	Herb	-	None	Cooked	-	-	[9]
48.	*Tulipa stellata* Hook.; Liliaceae; B.She.197.UOP	Ghantol	+	+	+	+	Fruit	Herb	-	Antiseptic, sinus pain	Cooked	11	0.142	No
49.	*Veronica polita* Fr.; Plantaginaceae; B.She.198.UOP	Angrara	-	+	+	-	Leaves	Herb	-	None	Cooked	-	-	No
50.	*Vicia sativa* L.; Fabaceae; B.She.199.UOP	Cheelo	+	+	-	-	Leaves and fruit	Herb	-	None	Eaten raw as a snack, salad	-	-	[48]
51.	*Zanthoxylum armatum* DC.; Rutaceae; B.She.200.UOP	Dambara	+	+	+	+	Fruit	Tree	$2.50	Asthma, bronchitis	Seasoning	75	0.721	No

UZ: Umarzai; HZ: Hadarzai; HK: Haji Khel; AK: Alisher Khel, UR: use report; mCI: cultural importance index.

**Table 4 biology-10-00537-t004:** Seasonal availability of wild vegetables in the Gadoon Valley.

S. No.	Species Name	Jan	Feb	Mar	Apr	May	Jun	Jul	Aug	Sep	Oct	Nov	Dec	Avail *
1.	*Allium griffithianum*													4
2.	*Allium jacquemontii*													2
3.	*Alternanthra sessilis*													7
4.	*Amaranthus spinosus*													8
5.	*Amaranthus viridis*													8
6.	*Apium graveolens*													6
7.	*Artemisia scoparia*													5
8.	*Asparagus officinalis*													4
9.	*Asphodelus tenuifolius*													3
10.	*Bauhinia variegate*													3
11.	*Berberis lycium*													3
12.	*Bidens pilosa*													8
13.	*Bistorta amplexicaulis*													4
14.	*Brassica carinata*													2
15.	*Capsella bursa-pastoris*													3
16.	*Caralluma tuberculata*													8
17.	*Chenopodium album*													4
18.	*Chenopodium murale*													8
19.	*Cichorium intybus*													7
20.	*Colocasia* spp.													4
21.	*Commelina benghalensis*													3
22.	*Convolvulus arvensis*													4
23.	*Eclipta prostrata*													3
24.	*Ficus palmata*													7
25.	*Lactuca serriola*													3
26.	*Lathyrus aphaca*													3
27.	*Lepidium apetalum*													4
28.	*Malva neglecta*													8
29.	*Medicago denticulata*													4
30.	*Medicago polymorpha*													6
31.	*Mentha longifolia*													8
32.	*Mentha spicata*													8
33.	*Nasturtium officinale*													8
34.	*Oxalis corniculata*													8
35.	*Pimpinella saxifraga*													2
36.	*Plantago lanceolata*													4
37.	*Portulaca oleracea*													4
38.	*Rumex dentatus*													8
39.	*Rumex hastatus*													8
40.	*Rumex vesicarius*													6
41.	*Silene conoidea*													4
42.	*Silene vulgaris*													4
43.	*Sisymbrium orientale*													4
44.	*Solanum nigrum*													6
45.	*Sonchus arvensis*													4
46.	*Trifolium repens*													4
47.	*Trigonella gracilis*													4
48.	*Tulipa stellata*													4
49.	*Veronica polita*													8
50.	*Vicia sativa*													3
51.	*Zanthoxylum armatum*													2

Avail *: number of months during which the plant is normally available.

## Data Availability

All the data is provided in the article.

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
