# Peer review of "Overcoming Tribal Boundaries: The Biocultural Heritage of Foraging and Cooking Wild Vegetables among Four Pathan Groups in the Gadoon Valley, NW Pakistan"

_biology, 2021, doi:10.3390/biology10060537_

Round 1

Reviewer 1 Report

Dear Colleagues,

First of all, I would like to congratulate you on your work, which has been very interesting to read.

I really enjoyed the ethnographic sections and I think it is pertinent to include data related to the informants. However, I felt that the methodology is not clearly explained. The final diagrams are very complete but I think it is necessary to revise a couple of texts and develop the explanation of point 2.6. Cultural Importance, and Significance Index.

References:

Shaheen, H., Qureshi, R., Qaseem, M. F., Amjad, M. S., & Bruschi, P. (2017). The cultural importance of indices: A comparative analysis based on the useful wild plants of Noorpur Thal Punjab, Pakistan. European Journal of Integrative Medicine, 12, 27–34. https://doi.org/10.1016/j.eujim.2017.04.003

Tardío, J., & Pardo-de-Santayana, M. (2008). Cultural Importance Indices: A Comparative Analysis Based on the Useful Wild Plants of Southern Cantabria (Northern Spain)1. Economic Botany, 62(1), 24–39. https://doi.org/10.1007/s12231-007-9004-5

English must be checked in some sections. Some words in the text have been marked.

On the other hand, it is not very convenient to conclude this paper with a reference to economic sovereignty, since it is not the main objective of this work, and the data that have been provided are not in tune with an economic analysis. They are small sentences that can be rephrased in a less categorical way, since their intentions are very positive.

I will be happy to review your work again after the reviewers' suggestions.

Thank you again for your efforts and sharing your work.

Author Response

The changes has been made as the worthy reviewer suggested

Reviewer 2 Report

The manuscript entitled " Overcoming tribal boundaries: The biocultural heritage of gathering and consuming wild vegetables among four Pathan groups in the Gadoon Valley, Swabi District, NW Pakistan ". Title, abstract and overall rationale of work to some extent is good. However, there are still some major concerns, which needs to be addressed and needs substantial revision.

  1. The content of the manuscript is too lengthy, specifically introduction section. Moreover, mechanism section is not properly elaborated, which is important for the reproducibility of the research.

  1. A flowchart should be added to this article to show the clear methodology.

  1. Author first time reported that, fifteen new wild vegetables from Khyber Pakhtunkhwa. However, author must need to explain all taxonomical classification of these plants and authors also need to justify how all these vegetable plant are significant for human welfare (specially medical purpose) and can authors characterized all these plants through NMR to know some important component belonging to these plants.

  1. I would suggest the authors to enhance your theoretical discussion and arrives your debate or argument.

  1. The resolution of figure 1 is not good and need to enhance 300 DPI.

  1. The number of interviewer in this study is very few and need to increase interviewer number.

  1. Authors must write the future prospective of this work and also explain in details significant of these all new 15 wild vegetable plants.

  1. Conclusion section must be elaborated,

Author Response

The revision has been made as suggested worth reviewer

Reviewer 3 Report

Dear editor, dear authors, 

The introduction is very nice and detailed written with a lot of references that is needed to mention latter in discussion.

My comments are mostly about improving the discussion, as well as improving the clarity of the text in some parts. 

I would suggest that the whole discussion be further supplemented by comparisons with  way of uses of well-known European plants such as Mentha, Chenopodium, Trifolium repens, Solanum nigrum ecc. in other areas of Pakistan and broader.

Species such as Silene vulgaris, Solanum nigrum and Sisymbrium orientale are commonly used thrue Mediterranean area, it's good to discuss why (reasons) and how are here mentioned for the first time in Pakistan?

I suggest comparing the habitat and altitude at which these newly recorded species occur, is it because some of them are alpine plants? But is it really for some edible uses of cosmopolitans to be recorded here for the first time?

I would also suggest that usage and application for newly recorded plants (for the first time here)  describe and compare in more detail.

Author Response

The changes has been made as suggested by worthy reviewer 

Reviewer 4 Report

Dear Authors,

I have carefully read your manuscript entitled "Overcoming tribal boundaries: The biocultural heritage of gathering and consuming wild vegetables among four Pathan groups" and I found it full of novelties in the ethnobotanical field and very important for the territory you studied.
It does, however, have numerous errors and mistakes that need to be properly corrected by you.

The background should be broadened, especially in the first part of Introduction section, by referring to concrete case studies in other parts of the world and in areas of the Asian continent itself, that is not strictly Pakistan.
All the scientific nomenclature must be deeply revised, checked, corrected, integrated and properly reported using the indications I have given you in several part of the manuscript.
In particular, I recommend you the following:

Please, update the botanical nomenclature using http://powo.science.kew.org/

All scientific names must be reported in full and complete with their authorship at the first mention in the text, in all figure and table captions and in the same figures and tables. In the other cases, you should report the scientific name in its abbreviated form (Bauhinia variegata L. / B. variegata). Moreover, several scientific names are bad written. Please, check carefully the whole manuscript: I highlighted and corrected only some of them.

In the second colum of Table 3, please, check the updated botanical nomenclature according with http://powo.science.kew.org/

Moreover, use commas to separate Taxon, botanical family and voucher code in each line.

Figure 10 is not correct. You indicated only the genus name of the taxa considered in your study: this is an error. Sorry, but you must report the scientific names in full.

Figure 11. All scientific names must be reported in full and complete with their authorship at the first mention in the text, in all figure and table captions and in the same figures and tables.

I don't find cited in the text  Supplementary Table 1.

After all these improvements, I think that your manuscript could continue in its publication process.

Best wishes.

Author Response

Our manuscript has reviewed by only three reviewers

Round 2

Reviewer 2 Report

The authors have addressed all the concerns raised in the previous version of the manuscript and the quality has improved after incorporating required modifications. Therefore, the manuscript may be considered for publication in this Journal.

Reviewer 3 Report

Dear autors and Editor, 

The name of author Sulaiman is missing in the text.
Key words must be diferrent than words in title, eg. edible plants, traditional knowledge erosion.

After your corrections and additions, I have no major objections. There are a few other small mistakes in the text, as well as repeated comments to which I did not receive an answer / explanation before.

In supplementary  materials Table 2 and 3 can be combined because they show the same data list.
